# Should homes and workplaces purchase portable air filters to reduce the transmission of SARS-CoV-2 and other respiratory infections? A systematic review

**Ashley Hammond** *, **Tanzeela Khalid, Hannah V. Thornton, Claire A. Woodall , Alastair D. Hay**

Centre for Academic Primary Care, NIHR School for Primary Care Research, Bristol Medical School: Population Health Sciences, University of Bristol, Bristol, United Kingdom

* ashley.hammond@bristol.ac.uk

**Data Availability Statement:** All relevant data are within the paper and its Supporting Information files.

## Abstract

Respiratory infections, including SARS-CoV-2, are spread via inhalation or ingestion of airborne pathogens. Airborne transmission is difficult to control, particularly indoors. Manufacturers of high efficiency particulate air (HEPA) filters claim they remove almost all small particles including airborne bacteria and viruses. This study investigates whether modern portable, commercially available air filters reduce the incidence of respiratory infections and/or remove bacteria and viruses from indoor air. We systematically searched Medline, Embase and Cochrane for studies published between January 2000 and September 2020. Studies were eligible for inclusion if they included a portable, commercially available air filter in any indoor setting including care homes, schools or healthcare settings, investigating either associations with incidence of respiratory infections or removal and/or capture of aerosolised bacteria and viruses from the air within the filters. Dual data screening and extraction with narrative synthesis. No studies were found investigating the effects of air filters on the incidence of respiratory infections. Two studies investigated bacterial capture within filters and bacterial load in indoor air. One reported higher numbers of viable bacteria in the HEPA filter than in floor dust samples. The other reported HEPA filtration combined with ultraviolet light reduced bacterial load in the air by 41% (sampling time not reported). Neither paper investigated effects on viruses. There is an important absence of evidence regarding the effectiveness of a potentially cost-efficient intervention for indoor transmission of respiratory infections, including SARS-CoV-2. Two studies provide 'proof of principle' that air filters can capture airborne bacteria in an indoor setting. Randomised controlled trials are urgently needed to investigate effects of portable HEPA filters on incidence of respiratory infections.

**Funding:** National Institute for Health Research (NIHR) Senior Investigator Award for ADH (NIHR NIHR200151). The funder of the study had no role in study design, data collection, data analysis, data interpretation, or writing of the report. The corresponding author had full access to all the data in the study and had final responsibility for the decision to submit for publication.

**Competing interests:** The authors have declared that no competing interests exist.

## Introduction

Respiratory infections such as coughs, colds, and influenza, are common in all age groups, and can be either viral or bacterial. Viral dimensions range between 0·02–0·3μm, and bacteria 0·5–10μm [1]. Bacteria and viruses can become airborne via talking, coughing or sneezing which generate aerosols (diameter up to 5μm) or droplets (diameter larger than 5μm) [2, 3]. Some of the most pathogenic viruses such as influenza, respiratory syncytial virus, adenovirus and coronavirus can be aerosolised [2]. There is also evidence that respiratory pathogens are carried in aerosols, known as bioaerosols [1], and that vomiting can aerosolise norovirus [4]. Once inhaled or swallowed, microbes may invade the respiratory or gastrointestinal mucosa, causing infection. Airborne particles may also land on surfaces and hands, increasing the chance of direct and indirect transmission.

The current global coronavirus (COVID-19) pandemic is spread primarily by airborne droplets [5], and to date has led to over one million deaths worldwide [6]. Controlling acquisition and transmission of respiratory infections is of huge importance, particularly within indoor environments such as care homes, households, schools/day care, office buildings and hospitals where people are in close contact [7]. Reducing airborne microbes could reduce respiratory as well as urinary, gastrointestinal and skin infections, transmitted via contamination of hands, fomites and close contact, by reducing the number of microbes that land onto surfaces [8].

High specification filters cleanse the air of aerosols and droplets [9]. Air filtration is commonly used to reduce infections in high-risk healthcare environments such as operating theatres (to reduce surgical site infections) [10], and hospital rooms for people with severe immunodeficiency. Portable air filtration units, initially developed to trap vehicle emission particulates and pollens larger than 0·02μm in diameter, are more than sufficient to capture droplet and aerosolised bacteria and viruses [1]. SARS-CoV-2 is approximately 0·1μm in diameter [11]. Some products now commercially available for domestic use contain high efficiency particulate air (HEPA) filters, and have been shown capable of removing >99% of aerosolised H1N1 influenza particles from a 28·5m$^3$ test chamber in 20 minutes [12].

Despite many years of use within healthcare environments such as operating theatres, it remains unclear what effect air filtration has when used in other settings, including care homes, schools, day cares and workplaces. Several manufacturers of portable filters have stated that their air purifiers have been tested in experimental chambers, which most often acts to replicate aerosolised particles. However, there is often no detailed evidence provided on their websites to corroborate their claims for potential consumers to review before purchasing (see Table 1). This systematic review and narrative synthesis aims to investigate whether portable filters used in any indoor setting can reduce incidence of respiratory infections and thus, whether there is any evidence to recommend their use in these settings to reduce the spread of SARS-CoV-2 and other respiratory infections. Further, this review will explore whether portable filters in indoor settings capture airborne bacteria and viruses, and if so, what specifically is captured.

## Materials and methods

### Search strategy and selection criteria

We searched Medline, Embase and Cochrane for articles published in any language between January 2000 and March 2021. Medline and Embase were searched using Ovid interrogation software. MeSH terms for these databases were broad, and included "air filters", "air microbiology" and "infection control". MeSH terms were combined with text word searches which included "filtration", "filter", "purifier", and "HEPA filter". Grey and unpublished literature

**Table 1. Air filter manufacturer websites and whether the claims made on the website are corroborated with detailed evidence.**

| Manufacturer (product) | Link to website | Claims | Link to evidence on website (Yes/ No) |
|---|---|---|---|
| Philips (Series 3000i) | Series 3000i Air Purifier | Removes >99.97% of ultrafine particles | No[a] |
| Dyson (Pure Cool) | Pure Cool Air Purifier | Captures >99.95% of ultrafine particles | No[b] |
| Blueair (Classic 405) | Classic 405 | Removes particles down to 0.1 microns | No |
| Airvia Medical (Pro 150) | Pro 150 Air Purifier | Filters at least 99.97% of particles with diameter ≥0.01 μm in a single pass | No |
| HoMedics (AR-29) | AR-29 Air Purifier | Removes up to 99.97% of airborne contaminants | No |
| Aerobiotix (Illuvia) | Illuvia HUAIRS | Removes >99.97% particles | No |
| Beurer (LR 200) | LR 200 Air Purifier | Filters 99.5% of particles ≥0.3 μm | No[c] |
| Winix (Zero Pro) | Zero Pro Air Purifier | All WINIX air purifiers feature a True HEPA filter that filters 99.97% of particulate matter, pollen and allergens so you can breathe healthy air and minimise the risk of colds. | No |
| Dimplex (XPAP6) | XPAP6 Air Purifier | Ultraviolet light destroys micro-organisms such as germs, viruses, bacteria and fungi (such as mould toxins). | No |
| Vax (Pure Air 200) | Pure Air 200 Air Purifier | Removes 99% of harmful particles as small as 0.3 microns | No |
| ISGfume (Viralair) | Air Filtration Systems | Our Viralair-HEPA™ product collects viruses and bacteria. Lower risk of cross contamination and spread of disease. | No |
| AeraMax (Pro) | AeraMax Pro Air Purifiers | Removes up to 99.99% of airborne contaminants | No[d] |
| IQAir (HealthPro series) | HealthPro Series Air Purifier | Filters 99.5% of ultrafine pollution particles down to 0.003μm (including viruses 0.005–0.03μm and bacteria 0.5–10μm) | No |
| GermGuardian AC4825 Air Purifier | AC4825 Air Purifier | HEPA filter captures 99.97% of dust and allergens down to 0.3μm and the UV-C light helps kill airborne viruses and bacteria | No |
| Rensair air purifier | Air Purifiers | Removes and kills up to 99.97% of viruses, bacteria, pollen, mould/yeast, dust, allergens and odours with use of 13 HEPA filters and UV-C light. Claims to remove bacteria and virus colonies to undetectable levels and removes the coronavirus family with 99.98% effectiveness. | Yes[e] |

[a] Website states the purifier test was conducted at Airmid Healthgroup Ltd. Tested in a 28.5-m3 test chamber contaminated with airborne influenza A(H1N1). No further details provided on their website.

[b] Website states internal testing conducted for filtration efficiency (EN1822) at 0.1 microns and whole room coverage (TM-003711 & DTM801) in a 27m² room. No further details provided on their website.

[c] Website states that various bacteria and viruses are filtered out of the air using a triple-layer filter system. No other evidence provided.

[d] Website states that AeraMax Pro purifiers are certified to be effective in reducing airborne concentrations of influenza A (H1N1) aerosol in a test chamber, reaching 99.9% airborne virus reduction within the first 35 minutes of operation; are certified to capture 99.97% of pollutants at 0.3 microns; can capture more than 97.8% of pollutants at 0.1–0.15 microns, via IBR Laboratories test data.

[e] Tested by three independent laboratories with reports available on their website. Eurofins conducted germ count measurements before and after filter use.

was searched for using ISI Web of Science software and included journal articles, patents, websites, conference proceedings, government and national reports and open access material. Reference lists of selected key papers were also screened. All full-text papers were subject to citation searches. See S1 Table for full search strategy. This systematic review is registered on PROSPERO CRD42020211235.

One reviewer (AH) screened all titles and abstracts, which were then subject to a 10% double-screen check by a second reviewer (TK). Population, Intervention, Comparator, Outcome and Study design (PICOS) criteria were used for inclusion and exclusion decisions, outlined in Table 2. Studies were eligible for inclusion if they included a portable air filter in any indoor setting including care homes, schools or healthcare settings, investigating either associations with incidence of respiratory infections or removal and/or capture of aerosolised bacteria and

**Table 2. Population, intervention, comparator, outcome and study design criteria for inclusion and exclusion.**

| PICOS | Inclusion criteria | Exclusion criteria |
|---|---|---|
| **Population** | Any population and age group | Outdoor settings |
| | Any country | Aircrafts |
| | Indoor community-setting, including but not limited to: | Specially designed 'germ free' chambers |
| | • Households | |
| | • Care homes | |
| | • Schools | |
| | • Nurseries/ day cares | |
| | • Universities | |
| | • Workplaces (offices) | |
| | • Public buildings | |
| | • Primary care practices | |
| | • Hospitals | |
| **Intervention** | Portable, commercially available air filters, including high efficiency particulate air (HEPA) filters | Non-portable air filters |
| | | Static, in-built filter systems used in hospital settings, as well as laminar air flow; turbulent mixing ventilation; |
| | | Positive/negative pressure systems |
| | | Experimentally designed filters (unlikely to be commercially available) |
| | | Air purifying respirators or face masks/personal protective equipment (PPE) |
| | | Filters that require forced air, fitted within an air duct. |
| **Comparator** | No air filter use within the same setting (for example randomised controlled trial of air filters in classrooms or offices) | |
| | Or not applicable if observational study | |
| **Outcomes** | Studies reporting effects of portable air filters on incidence of respiratory infection. | Studies which do not report effects of portable air filters on incidence of respiratory infections, and do not report which aerosolised bacteria and viruses are captured by the filter. |
| | Studies reporting whether filters capture/remove aerosolised bacteria and viruses from the air, including information of what is captured. | |
| **Study design** | Studies published after 2000 | Studies published before 2000 |
| | Randomised controlled trials | Qualitative studies without any quantitative data |
| | Non-randomised trials | |
| | Interventions | |
| | Observational studies | Systematic reviews |
| | • Cohort | Economic studies |
| | • Case-control | Critical reviews/expert opinions without any primary data |
| | • Cross-sectional | |
| | • Longitudinal | |
| | Epidemiological studies | |

viruses from the air within the filters. We chose to exclude any studies using non-portable filters, including those on aircrafts, and in some hospital or healthcare settings, which use static systems such as laminar airflow or positive or negative pressure systems.

## Data extraction and quality assessment

Full text papers for all eligible studies were obtained and data extracted by two independent reviewers (AH, TK, HT, CW) using a purpose-built spreadsheet. Where provided, the following information was extracted for all papers: author, journal, year of publication, study design, study country, setting and recruitment, study time period and filter used. Further analytical data was extracted from all papers on an individual basis.

For all observational studies, the Critical Appraisal Skills Programme (CASP) checklist was used to assess study quality (www.casp-uk.net). Our key quality criteria for eligibility were clear reporting of filters as being portable, placement of filters within a community setting, and clear reporting of methods and analysis. A risk of bias score of "high", "medium" or "low" was applied to each criterion.

## Synthesis

A narrative synthesis was performed to summarise the findings from different studies. We chose this method due to the differences in methods and reporting of the few studies that were included in this systematic review. Our first preliminary synthesis included a thematic analysis involving searching of studies, listing and presenting results in tabular form. The results were then discussed again with all authors, then summarised in a narrative synthesis within a framework by one author.

This framework consisted of the following factors: the filter used (eg. HEPA), the setting (eg. nursing homes, emergency rooms, offices, schools and day cares), the design/methods (eg. randomised controlled trial, observational), and how effective the filters were at either capturing aerosolised bacteria and viruses or removing/reducing them from the filtered air. These themes were discussed in relation to whether the study explored incidence of infection or removing/capturing airborne bacteria and viruses from the air. All articles that were included in this review were published, and all methods in this review were undertaken according to PRISMA guidelines [13].

## Results

Our search returned 15,750 papers (Fig 1). Following review of titles and abstracts 15,651 papers were excluded. The abstracts of 99 papers were reviewed, 81 papers were excluded, leaving 18 full texts to be reviewed (see S2 Table for a summary of eligibility criteria for these 18 studies). Of these, 16 were excluded for the following reasons: ten studies did not use portable filters, three studies were conducted in experimental germ-free chambers only

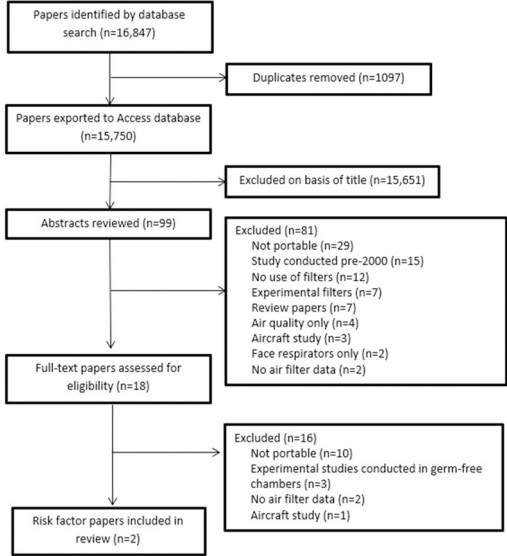

**Fig 1. Data search and extraction (PRISMA flow-chart).**

and not identified as commercially available filters, two studies did not report any data on the functionality of the air filters, and one study was conducted in an aircraft. This left two studies included in the review. One included study was conducted in Beijing in 2020 within an office setting [14]. The other study was conducted in the USA in 2019 in an emergency room [15]. Study quality was generally good and indicated low risk of bias overall, however neither study took into account any potential confounding factors, nor did they clearly indicate their study recruitment processes, i.e. how many offices or emergency rooms were invited to take part in the studies or clearly report all aspects of filter use methods such as placement of filter in the room or duration of use. Both studies had clear aims, summarised their results in line with methods, and conclusions appeared to be supported by the results. See Fig 2 for data quality chart.

## Studies exploring effects of portable air filters on incidence of respiratory infection in the community

We did not find any studies which investigated the effects of portable, commercially available air filters on incidence of respiratory infections. Further, our search strategy found only one study which investigated the effects of non-portable HEPA air filtration on incidence of pneumonia in severely immunocompromised patients [16].

## Studies investigating whether portable filters placed in an indoor setting capture airborne bacteria and viruses from the air

Our search identified two papers which investigated whether portable filters placed in an indoor setting capture and/or reduce airborne bacteria, but not viruses, from the air.

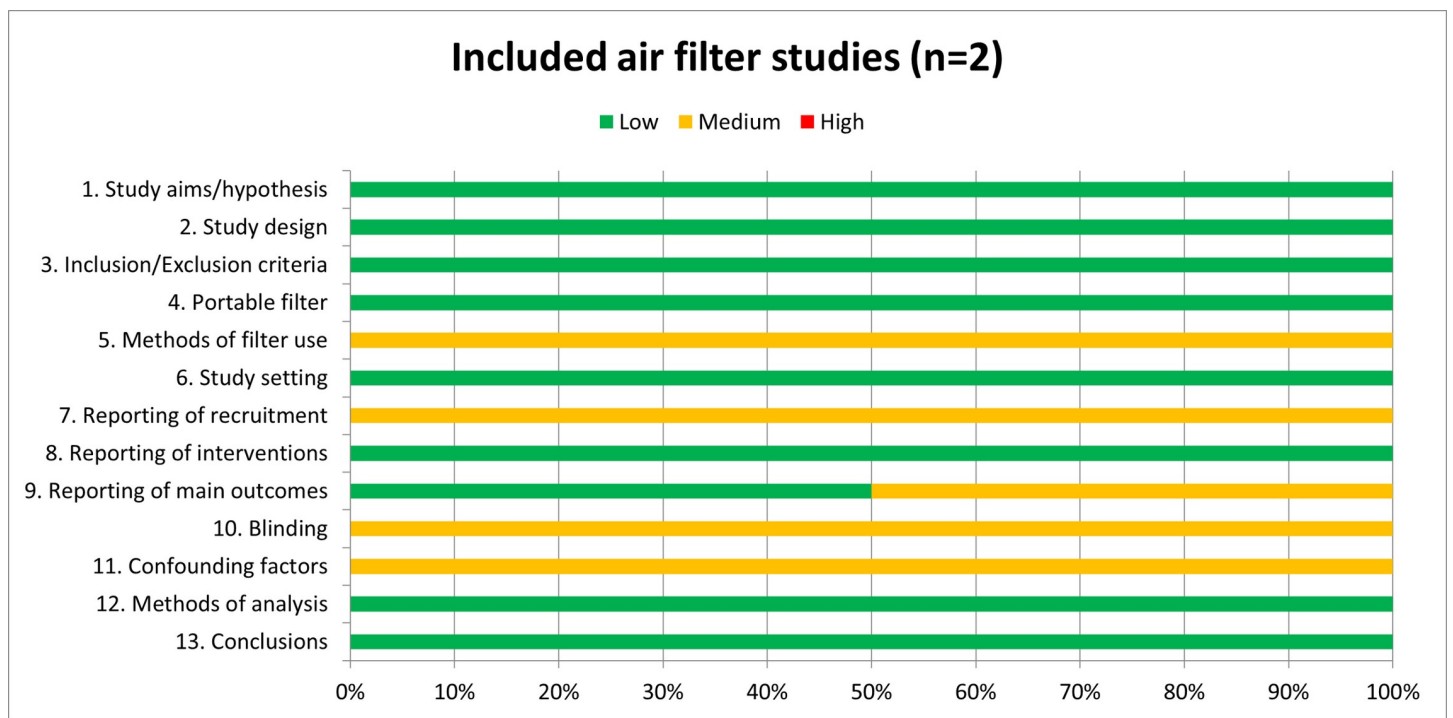

**Fig 2. Data quality charts based on CASP checklist.**

## Setting and methods

One observational study, conducted between March 2016 and March 2017 in Beijing, placed air purifiers inside 12 independent administrative offices in three buildings. The air purifiers were fitted with new HEPA filters and one placed per office for one year. Two dust sampling sites were marked on the floor in each office and were left undisturbed for the one-year study period. Sterile cotton swabs and face masks (for protection during sample collection) were used to collect floor dust samples, and a 2x2cm square of outer membrane was cut from each filter. The researchers also took samples of indoor and outdoor air (capturing bacteria on blood agar containing Petri dishes), human oral samples (using deionised water mouth-washes), outdoor air (using an airborne bacteria sampler), and soil (topsoil sample; 5 cm collected with disposable sterile shovel, with 20g soil inserted into sterile 50 ml tube) at various timepoints (indoor/outdoor air and human oral collected in December 2016; outdoor air and soil collected in September 2017). Fluorescent stains were used to determine cell membrane integrity for bacterial viability. Microbiome analysis (16S rRNA) was used to determine the type of bacteria trapped in the dust, air and filter samples.

Another observational study conducted in the USA, assessed the effectiveness of a portable filter in eliminating bacterial aerosols from emergency rooms [15]. A twenty-minute baseline air sample was taken using blood agar plates near the head and foot of patients' beds and at the doorway of patients' rooms before operation of the filter. A high-efficiency particulate air-ultraviolet air recirculation system (HUAIRS) was run inside the patients' rooms for eight air exchanges (washout phase, adjusted by room size) and air sampling was repeated for 20 minutes as before while the HUAIRS system was left on. The total amount of time the filters were switched on was not reported, nor was the time between sampling. The times of door openings were recorded during both measurement periods to assess possible impact on air burden. The level of bacterial growth was recorded as colony forming units (CFU) after 48 hours.

**Type of air filtration system used.** The Beijing study did not provide specific details of the filter they used, nor the smallest particle size the filter was designed to capture. They did provide a photograph, which indicated the air purifier with HEPA filter was small and evidently portable. Air purifier use ranged from 121 to 143 days.

The USA study used the Aerobiotix Illuvia 500uv system (Aerobiotix, West Carrollton, OH). The authors do not report any details on filter specification, in particular related to the smallest particle size the filter was designed to capture, nor could we find this information on the manufacturer website (https://aerobiotix.com/products/illuvia-500uv-2/). The website describes it as an "innovative, high-volume air purification system combining HEPA filtration, zirconium-based photochemical oxidation, and germicidal UV irradiation, targeting particulates, aerosol pathogens, and volatile organic compounds". The system is described as portable with a small footprint and low noise, making it suitable for use in indoor environments.

**Capture or removal/reduction of airborne bacteria.** The Beijing study found the survival rate of bacteria in filter samples were significantly higher than those in the dust samples. Significant differences between the taxonomic abundance and microbial composition of the filter and dust samples were determined by beta diversity measures Bray-Curtis dissimilarity, Jaccard distance and Unifrac. These analyses provide information on the difference in taxonomic abundance profiles from different the samples. The major classes in the filter samples were Alphaproteobacteria (51·8%) and Actinobacteria (17·2%), whereas those in dust samples were Bacteroidia (25·6%), Clostridia (13·9%), Bacilli (15·9%), Gammaproteobacteria (11·7%) and Alphaproteobacteria (11·3%). The filter and dust samples showed significant differences in both taxa at genus level. The major genera in filter samples were Sphingomonas (4·7%), Rubellimicrobium (3·5%) and Pseudonocardia (3·1%), whereas those in dust samples were

Streptococcus (11·2%) and Pantoea (3·9%). Furthermore, the number of unique operational taxonomic units (OTUs) in the HEPA filter samples (n = 738) was more than double that in the dust samples (n = 253) (Chao1, Shannon diversity index and phylogenetic diversity index).

HEPA filter samples had higher proportion of viable bacteria than in the dust samples. The reason for this is likely due to the filter trapping particles which provide enrichment for the filter-immobilised bacteria. Given that the study found significantly different bacterial communities and bacterial diversity between filter and dust samples, the authors conclude that the HEPA filter should represent a new ecological niche within indoor environments. The key sources of bacteria were soil for the HEPA filter, and human oral, indoor and outdoor air for the dust samples. No significant difference was found between the offices (p = 0·50).

For the USA emergency room study, the use of the HUAIRS system led to a 41% reduction in the mean CFU of aerosol bacterial load compared to before HUARIS use (using paired t-tests) for all particle sizes (p<0·05). When particle sizes were grouped into less or greater than 4·7μm, there was a significant reduction in bacterial burden from use of the HUAIRS system for the foot and doorway locations but not the head location (>4·7μm). The reasoning for choosing this particle size cut off was not reported. Door openings did not change the bacterial burden during baseline (p = 0·85) and HUAIRS runs (p = 0·32). Colony counts on blood agar were performed however no further species identification was carried out. There was no reporting of aerosol particle reduction to determine the effectiveness of the HUAIRS system.

## Discussion

We found no studies investigating the effects of portable, commercially available air filters on the incidence of respiratory infections in the community. Two papers reported removal or capture of airborne bacteria in indoor settings (office and emergency room), demonstrating that the filters captured airborne bacteria and reduced the amount of airborne bacteria in the air. Neither tested for the presence of viruses in the filters, nor a reduction in viral particles in the air.

To our knowledge, this is the first systematic review to explore the effects of portable, commercially available air filters on incidence of respiratory infections, and whether they capture airborne bacteria and viruses from the air within indoor settings. Despite finding very few studies, we adopted a systematic approach with a broad search strategy, and are confident we captured all evidence regarding modern portable HEPA filters currently available, including studies related to SARS-CoV-2, of which there were none. However, our review was restricted to commercially available portable filters; this was so that we could explore the effectiveness of air filters available for purchase and use in a 'real-world' setting. We therefore do not report the effects of non-portable systems like laminar airflow and positive and negative pressure systems, applicable in specialised environments such as aircraft and hospital operating theatres. We have also not explored the effects of within-building filtered air flow systems, common in offices in the UK, and in domestic settings worldwide, such as HVAC (heating, ventilation, and air conditioning) systems. We acknowledge that our eligibility criteria found only two studies, however we believe this only further highlights the considerable gap in evidence related to the effectiveness of portable air filters in preventing respiratory infections, including SARS-CoV-2.

The two included papers focused only on capture of bacteria in the air filters, as opposed to, or including viruses. As bacteria are larger in size than viruses, one might question whether these filters are also capturing viral particles. As there exists a complex relationship between bacteria and viruses, rarely do they exist as separate entities [17]. It is likely that if the air filter is capturing bacteria, it will also be capturing viral particles. As we found no papers which tested for both, we believe this demonstrates a further need for evidence on the effectiveness of air filters in capturing both bacterial and viral particles, and preventing the infections they can cause.

There is a paucity of evidence regarding the effectiveness of portable air filters in reducing incidence of respiratory infections, including SARS-CoV-2, in indoor environments. Our search returned only one study investigating incidence of infection but was not eligible for inclusion because the HEPA filters were built into the rooms [16]. This study compared outcomes for severely immunocompromised patients with and without HEPA-filtration. Pneumonia incidence in HEPA-filtered rooms was 7% (18/254) and 17% (6/35) in non-HEPA-filtered rooms (p = 0·05). There were no differences in mortality at 100 days: 14% vs. 17% (p = 0·6). The authors conclude that HEPA-filtered rooms should be used for these patients.

One study conducted in the USA investigated the effectiveness of portable HEPA filters in an empty plastic hospital anteroom on the spread of aerosolised particles, measuring the concentration of the 0·3μm, 1·0μm, and 3·0μm particles, with an emphasis on 0·3μm, as this is the closest to the size range of SARS-CoV-2 virus [18]. Respectively, these represent fine, intermediate, and large particle sizes, and cover a range from virus nuclei to viruses carried on droplets. When combined with a portable HEPA air purifier, aerosol containment within the anteroom was >99% (compared to 98% without the filter). These results do not suggest that the HEPA filter offered greater containment than the plastic anteroom itself, but where such rooms cannot be used still require further investigation to assess their effectiveness in reducing respiratory infection acquisition, including SARS-CoV-2.

Although non-portable air filtration systems were not included in this study, they can provide valuable information on the effectiveness of air filters in capturing airborne bacteria or viruses. One study conducted in the USA in 2011 sampled non-portable heating, ventilation and air conditioning (HVAC) filters within two public buildings for the presence of viruses, including coronaviruses [19]. Of the 64 filters tested, nine were positive for influenza A, two for influenza B, and one positive for parainfluenza virus 1.

Some commercially available air filters have been tested within experimental chambers, designed to replicate bioaerosols in an indoor setting. One study conducted in Canada in 2018 replicated specific bioaerosols to determine the effectiveness of a filter in reducing surface contamination [20]. The experimentally contaminated air was filtered using a portable device which combined HEPA filtration and UV light. After a 45-minute run, viable levels of tested bacteria in the air were reduced by >99%, and surface contamination was reduced by at least 87%. The combination of UV light alongside HEPA filters ensures the captured organisms are killed. Our included Beijing study highlighted that their HEPA filters had higher levels of viable bacteria than in the settled floor dust. Whilst experimental chamber studies provide some degree of evidence with respect to the filter specifications, they do not provide 'real-world' evidence of their effectiveness, including the specific airborne bacteria and viruses captured from the air and its overall effect on incidence of respiratory infections.

The UK Chartered Institution of Building Services Engineers (CIBSE) advise the UK government of appropriate air conditioning and ventilation during the pandemic. The CIBSE states that in poorly ventilated spaces with a high occupancy it could be appropriate to consider using air cleaning device. They advise the most appropriate devices would likely be local HEPA filtration units or those that use germicidal UV (GUV) radiation, but acknowledge that there is not yet any specific evidence of the effectiveness of UV-C irradiation for SARS-CoV-2 [21]. The United States Environmental Protection Agency states that air purifiers and HVAC filters can help to reduce aerosolised bacteria and viruses, including SARS-CoV-2, when used in combination with other recommended best practices such as social distancing, handwashing and surface disinfection [22]. While we found evidence to suggest the use of air filters could theoretically contribute to reducing the spread of SARS-CoV-2 and other respiratory infections by capturing the relevant airborne particles, there is a complete absence of evidence as to whether they actually reduce the acquisition of these infections. We consider there is

sufficient 'proof of principle' evidence to support the conduct of a randomised controlled trial to investigate this question. Further research is also needed regarding which types of air filter are most effective, and whether they should include filter germicidal capabilities.

Portable HEPA air filtration systems can reduce levels of airborne bacteria, but there is an absence of evidence regarding the removal of airborne viruses and there have been no studies investigating the effects of commercially available, portable air filters on incidence of SARS-CoV-2 or other respiratory infections in indoor settings. Randomised controlled trials are urgently needed to demonstrate the effects of portable HEPA air filters on incidence of respiratory infections, including those caused by SARS-CoV-2. The main research questions must focus primarily on whether use of portable HEPA filters in any indoor environment reduce respiratory infections compared to those environments without portable HEPA filters. In order to fully address whether there is value in homes and workplaces purchasing portable air filters to reduce respiratory infections, future research also needs to understand their effectiveness within different indoor environments, including large open-plan offices, care homes, nurseries and private homes. Cost-benefit analysis must also be conducted to understand whether the benefits of portable air filters in reducing respiratory infections outweigh the costs of purchasing the filters. In addition, qualitative analyses should be conducted to gain insight around uptake of portable air filters, in particular whether people would be prepared to purchase and use them.

## Conclusions

Our systematic review uncovered a considerable gap in evidence around whether portable air filters reduce the incidence of respiratory infections, including SARS-CoV-2. We reported findings from two studies, which both reported removal or capture of airborne bacteria only in indoor settings, and demonstrated that the portable filters did capture airborne bacteria and reduced the amount of airborne bacteria in the air. We did not find any studies investigating the effects of portable, commercially available air filters on the incidence of respiratory infections in the community. Governments worldwide continue to advise maintaining good ventilation in order to prevent the spread of SARS-CoV-2 and other respiratory infections, and portable air filters in theory provide a real-world solution for many indoor environments, however further research is urgently needed to assess their effectiveness in reducing the incidence of respiratory infections.

## Supporting information

**S1 Table. Medline search strategy.**
(DOCX)

**S2 Table. Summary of studies based on eligibility criteria and author best judgement (non-exhaustive list).**
(DOCX)

**S1 Checklist.**
(DOCX)

## Author Contributions

**Conceptualization:** Alastair D. Hay.

**Data curation:** Ashley Hammond, Hannah V. Thornton.

**Formal analysis:** Ashley Hammond, Tanzeela Khalid, Hannah V. Thornton, Claire A. Woodall, Alastair D. Hay.

**Funding acquisition:** Alastair D. Hay.

**Investigation:** Ashley Hammond, Tanzeela Khalid, Alastair D. Hay.

**Methodology:** Ashley Hammond, Tanzeela Khalid, Hannah V. Thornton, Claire A. Woodall, Alastair D. Hay.

**Project administration:** Ashley Hammond.

**Resources:** Ashley Hammond.

**Supervision:** Ashley Hammond, Alastair D. Hay.

**Validation:** Ashley Hammond, Tanzeela Khalid, Hannah V. Thornton, Claire A. Woodall.

**Writing – original draft:** Ashley Hammond, Tanzeela Khalid.

**Writing – review & editing:** Ashley Hammond, Tanzeela Khalid, Hannah V. Thornton, Claire A. Woodall, Alastair D. Hay.

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
