## [Decision Letter · Decision Letter 0]

17 Dec 2020

PONE-D-20-34977

Should homes and workplaces purchase portable air filters to reduce the transmission of SARS-CoV-2 and other respiratory infections? A systematic review

PLOS ONE

Dear Dr. Hammond,

Thank you for submitting your manuscript to PLOS ONE. After careful consideration, we feel that it has merit but does not fully meet PLOS ONE’s publication criteria as it currently stands. Therefore, we invite you to submit a revised version of the manuscript that addresses the points raised during the review process.

A systematic review is a review of a clearly formulated question that uses systematic and reproducible methods to identify, select and critically appraise all relevant research, and to collect and analyze data from the studies that are included in the review. Overall, your approach has merit, but the aspects pointed out by the reviewers should be improved. After reading the manuscript, I suggest to increase the complexity of the presentation by shifting tables S1 and S3 and figure S1 in the main text. Furthermore, you need to provide a table presenting the summary of selected studies (n= 18) based on eligibility criteria and best judgment and point out that it could be a non-exhaustive list since a degree of subjectivity may be present (see an example in supplementary of https://www.mdpi.com/2073-4433/9/4/150/htm - table S2). A better clarification of the database selection should be also included (one may ask why Clarivate WoS was not consulted?).

We look forward to receiving your revised manuscript.

Kind regards,

Daniel Dunea, Ph.D.

Academic Editor

PLOS ONE

Journal Requirements:

Reviewers' comments:

Reviewer's Responses to Questions

**Comments to the Author**

1. Is the manuscript technically sound, and do the data support the conclusions?

Reviewer #1: Partly

Reviewer #2: Partly

2. Has the statistical analysis been performed appropriately and rigorously? 

Reviewer #1: N/A

Reviewer #2: N/A

3. Have the authors made all data underlying the findings in their manuscript fully available?

Reviewer #1: Yes

Reviewer #2: Yes

4. Is the manuscript presented in an intelligible fashion and written in standard English?

Reviewer #1: Yes

Reviewer #2: Yes

5. Review Comments to the Author

Reviewer #1: I want to congratulate authors for completing this manuscript. This manuscript reviewed literatures regarding whether using portable air filters can reduce the transmission of respiratory disease pathogens among people, including SARS-CoV-2, the virus which are responsible for the on-going pandemic. This is a well-written manuscript and address using air filters as an intervention strategy to reduce coronavirus disease 2019 (COVID-19) and/or other respiratory infection. However, I felt there are still some weakness of this manuscript, and there is main issue of the significance contribution of the manuscript. I would like to recommend this manuscript to be published given my comments and concerns are adequately addressed, and the importance of the findings of this manuscript are improved.

Major comments:

1. Limited number of studies. Based on authors’ search strategy, authors did not found any study investigating the effects of portable, commercially available air filters on incidence of respiratory infections. This review only found two papers which investigated whether portable air filters can reduce airborne bacteria from the air. These very limited number of studies reduce the significance of this manuscript. I would recommend authors to expand research criteria (such as including centralized HVAC system with air filter studies etc.) to include more studies. Expanding beyond portable air filter studies would provide important evidence of the efficacy of using other air filtration technology in reducing transmission of COVID-19.

2. The two studies included in the systematic review only focused on air filter use on bacterial in the air, but not virus. Given the different size and pathological characteristics between bacteria and virus, I would encourage authors add some more discussions on how the results from bacteria can inform controlling virus.

3. In this manuscript, there is only 1 figure, but no tables. I would like encourage authors to include 1-2 tables summarizing two studies they included in the synthesis.

Reviewer #2: This is a well conducted systematic review for an important question particularly in the context of the current COVID-19 pandemic. The main objective of this review is to investigate whether commercially available portable air filters used in a ‘real-world’ indoor setting can reduce incidence of respiratory infections and whether these filters placed indoors can capture airborne bacteria and viruses. To answer this question, the author conducted a systematic review and developed criteria that included two types of studies: 1) studies exploring the effects of portable air filters on incidence of respiratory infection in the community and 2) studies examining whether portable air filters in an indoor setting capture airborne bacteria and viruses from the air. The author found no studies investigating the effects of portable, commercially available air filters on the incidence of respiratory infections in community and two studies showed that the filters captured airborne bacteria and reduced the number of airborne bacteria in the air, but no evidence for viruses.

The manuscript is certainly well written and attractive. The method and quality control are also solid. But given the author’s review scope and inclusion criteria only landed two studies with limited comparability, I’m concerned that this review may not provide sufficient information that qualifies a publication.

The included two studies are largely different in indoor settings (office vs. emergency room), sampling duration (one year vs. one day?), air filter type (air purifier with HEPA filter vs. high-efficiency particulate air-ultraviolet air recirculation system). Additionally, the missing information (e.g., filter specifications, filter use duration/frequency as well as the air exchange of the indoor environment etc.) makes the author’s claim on the ability of air filters to capture airborne bacteria less convincing. The author may want to consider expanding the scope of this review to also include non-portable HEPA filters. As the author recognized, the within-building filtered air flow systems such as HVAC systems are commonly equipped in office and domestic settings in UK and worldwide. It is certainly an important ‘real-world’ setting as well. Including this body of literature may provide additional information and evidence on the effectiveness of air filters in reducing respiratory infections or capturing airborne bacteria or viruses. It may also help answer other practical questions such as should we recommend people who stay in the indoor environment with within-building air filtration to acquire additional portable air filters?

I really like that the author also summarized literatures that do not fit into the review scope/inclusion criteria but relevant to the topic in the discussion section. It provides a great overview of the existing literature on this topic. But for the discussion on future research, I would like to see more details rather than an oversimplified statement that randomized controlled trials are urgently needed. What are the research questions need to be answered? What evidences are needed to make concrete recommendations and what are the immediate challenges to overcome?

In summary, this is a useful review trying to answer important questions, but it could have been improved by providing more information and insights to tell a more complete story.

Minor issues:

1. Line 59 – 62: References?

2. Line 64: References?

3. Line 67 – 69: Not clear why this study is cited here. It is not related to air filters as the sentence before and after. Perhaps a reference to the statement in line 59 - 62?

4. The review seems to focus on effectiveness, but the one included study was assessing the efficacy. The authors may want to be clear about the scope and make sure “effectiveness” and “efficacy” are used carefully and consistent throughout the manuscript.

6. PLOS authors have the option to publish the peer review history of their article (what does this mean?). If published, this will include your full peer review and any attached files.

Reviewer #1: No

Reviewer #2: No

---

## [Author Response · Author response to Decision Letter 0]

19 Mar 2021

We have included a table detailing our response to all editor and reviewer comments in our cover letter included with this revision.

---

## [Decision Letter · Decision Letter 1]

14 Apr 2021

PONE-D-20-34977R1

Should homes and workplaces purchase portable air filters to reduce the transmission of SARS-CoV-2 and other respiratory infections? A systematic review

PLOS ONE

Dear Dr. Hammond,

Thank you for submitting your manuscript to PLOS ONE. After careful consideration, we feel that it has merit but does not fully meet PLOS ONE’s publication criteria as it currently stands. Therefore, we invite you to submit a revised version of the manuscript that addresses the points raised during the review process.

After consulting the revised manuscript and reviewers' comments, I recommend minor revision because I believe that the manuscript would be more conclusive if the authors would include in the discussion the limitations of the study by answering the problems raised by both reviewers and their own opinions on the potential limitations. Furthermore, the authors should make suggestions for further research especially because they found few papers on this topic. They should include also a Conclusion section.

Consequently, please address these issues before re-submission:

- Acknowledge the review's limitations

- Make Suggestions for Further Research

For these two points consider the comments of both reviewers:

- "elaborate more on the future directions of research (e.g. specific indoor settings, population, cost-benefit, uptake/adoption and other practical implications) to better reflect the manuscript title: “Should homes and workplaces purchase portable air filters…”, as well as the focus of effectiveness. "

- "citing 2 papers in a review is relatively weak."

Although your study may offer important insights about the research problem, other questions related to the problem likely remain unanswered. Moreover, some unanswered questions may have become more focused because of your study. You should make suggestions for further research in the discussion section.

- include a Conclusion section synthesizing the key findings and ideas.

A well-written conclusion provides you with several important opportunities to demonstrate your overall understanding of the research problem to the reader.

The function of your review's conclusion is to restate the main argument. It reminds the reader of the strengths of your main argument and reiterates the most important evidence supporting that argument. Make sure, however, that your conclusion is not simply a repetitive summary of the findings because this reduces the impact of the arguments you have developed so far.

Summarize briefly the key findings and restate a key statistic, fact, or strong idea to drive home the ultimate point of your review (you can use a bulleted list as well).

We look forward to receiving your revised manuscript.

Kind regards,

Daniel Dunea, Ph.D.

Academic Editor

PLOS ONE

Journal Requirements:

Reviewers' comments:

Reviewer's Responses to Questions

**Comments to the Author**

1. If the authors have adequately addressed your comments raised in a previous round of review and you feel that this manuscript is now acceptable for publication, you may indicate that here to bypass the “Comments to the Author” section, enter your conflict of interest statement in the “Confidential to Editor” section, and submit your "Accept" recommendation.

Reviewer #1: All comments have been addressed

Reviewer #2: All comments have been addressed

2. Is the manuscript technically sound, and do the data support the conclusions?

Reviewer #1: Partly

Reviewer #2: Yes

3. Has the statistical analysis been performed appropriately and rigorously? 

Reviewer #1: N/A

Reviewer #2: N/A

4. Have the authors made all data underlying the findings in their manuscript fully available?

Reviewer #1: Yes

Reviewer #2: Yes

5. Is the manuscript presented in an intelligible fashion and written in standard English?

Reviewer #1: Yes

Reviewer #2: (No Response)

6. Review Comments to the Author

Reviewer #1: Thanks for including me to review this manuscript. I am glad that authors addressed my previous comments, even though I still think citing 2 papers in a review is relatively weak. Authors made strong argument for publishing their manuscript even with 2 papers, and I tentatively agree with that. Therefore, I recommend accepting this manuscript and publish it in the journal PLOS One as soon as possible.

Reviewer #2: The authors have clarified the major concern I raised in my previous review regarding the scope of the review and I respect the authors’s decision to conduct a separate review focusing on the effectiveness of non-portable air filters. The authors have also updated the manuscript and addressed the minor issues. I would again recommend the authors to elaborate more on the future directions of research (e.g. specific indoor settings, population, cost-benefit, uptake/adoption and other practical implications) to better reflect the manuscript title: “Should homes and workplaces purchase portable air filters…”, as well as the focus of effectiveness. Other than that, I do not have any further comments and approve this manuscript to be accepted.

7. PLOS authors have the option to publish the peer review history of their article (what does this mean?). If published, this will include your full peer review and any attached files.

Reviewer #1: No

Reviewer #2: No

---

## [Author Response · Author response to Decision Letter 1]

15 Apr 2021

We have included within our cover letter attached a table detailing our responses to all editor and reviewer comments.

---

## [Editor Report · Decision Letter 2]

20 Apr 2021

Should homes and workplaces purchase portable air filters to reduce the transmission of SARS-CoV-2 and other respiratory infections? A systematic review

PONE-D-20-34977R2

Dear Dr. Hammond,

We’re pleased to inform you that your manuscript has been judged scientifically suitable for publication and will be formally accepted for publication once it meets all outstanding technical requirements.

Kind regards,

Daniel Dunea, Ph.D.

Academic Editor

PLOS ONE

---

## [Editor Report · Acceptance letter]

22 Apr 2021

PONE-D-20-34977R2 

Should homes and workplaces purchase portable air filters to reduce the transmission of SARS-CoV-2 and other respiratory infections? A systematic review 

Dear Dr. Hammond:

I'm pleased to inform you that your manuscript has been deemed suitable for publication in PLOS ONE. Congratulations! Your manuscript is now with our production department. 

Kind regards, 

on behalf of

Prof. Daniel Dunea 

Academic Editor

PLOS ONE